# Stratification of hypertension and SARS-CoV-2 infection by quantitative NMR spectroscopy of human blood serum

Jasmin Kazenwadel[1], Georgy Berezhnoy[1], Claire Cannet[2], Hartmut Schäfer[2], Tobias Geisler[3], Anne-Katrin Rohlfing[3], Meinrad Gawaz [3], Uta Merle [4,5] & Christoph Trautwein [1,5✉]

**Abstract**

**Background** Diagnostic approaches like the nuclear magnetic resonance spectroscopy (NMR) based quantification of metabolites, lipoproteins, and inflammation markers has helped to identify typical alterations in the blood serum of COVID-19 patients. However, confounders such as sex, and comorbidities, which strongly influence the metabolome, were often not considered. Therefore, the aim of this NMR study was to consider sex, as well as arterial hypertension (AHT), when investigating COVID-19-positive serum samples in a large age-and sex matched cohort.

**Methods** NMR serum data from 329 COVID-19 patients were compared with 305 healthy controls. 134 COVID-19 patients were affected by AHT. These were analyzed together with NMR data from 58 hypertensives without COVID-19. In addition to metabolite, lipoprotein, and glycoprotein data from NMR, common laboratory parameters were considered. Sex was considered in detail for all comparisons.

**Results** Here, we show that several differences emerge from previous NMR COVID-19 studies when AHT is considered. Especially, the previously described triglyceride-rich lipoprotein profile is no longer observed in COVID-19 patients, nor an increase in ketone bodies. Further alterations are a decrease in glutamine, leucine, isoleucine, and lysine, citric acid, HDL-4 particles, and total cholesterol. Additionally, hypertensive COVID-19 patients show higher inflammatory NMR parameters than normotensive patients.

**Conclusions** We present a more precise picture of COVID-19 blood serum parameters. Accordingly, considering sex and comorbidities should be included in future metabolomics studies for improved and refined patient stratification. Due to metabolic similarities with other viral infections, these results can be applied to other respiratory diseases in the future.

**Plain language summary**

The functionality of our human body is driven by a large number of small molecules, called metabolites. These metabolites can be associated with health but also disease conditions. In this study, we used a technology called nuclear magnetic resonance spectroscopy (NMR) to determine metabolite and protein concentrations in the blood of acutely-infected COVID-19 patients and compared these results with disease severity and clinical laboratory data. We particularly focus on patients with the very common cardiovascular condition, arterial hypertension (AHT), and important factors such as sex, age and medication. Our findings provide a more detailed insight into COVID-19 and which individuals are at higher risk for more severe disease.

[1] Werner Siemens Imaging Center, Department for Preclinical Imaging and Radiopharmacy, Eberhard Karls University Tübingen, Tübingen, Germany. [2] Bruker BioSpin GmbH, Applied Industrial and Clinical Division, Ettlingen, Germany. [3] Department of Internal Medicine III, Cardiology and Angiology, University Hospital Tübingen, Tübingen, Germany. [4] Department of Internal Medicine IV, University Hospital Heidelberg, Heidelberg, Germany. [5] These authors contributed equally: Uta Merle, Christoph Trautwein. ✉email: Christoph.Trautwein@med.uni-tuebingen.de

To date, the SARS-CoV-2 virus, which emerged in Wuhan, China in December 2019, has infected more than 760 million people worldwide and caused 6.9 million deaths from December 2019 to June 2023 (WHO). SARS-CoV-2 infections can manifest in a wide range of symptoms with varying disease severity. Patients often report mild flu-like symptoms or asymptomatic courses. In some cases, however, intensive medical treatment may be necessary, or the infection may be fatal. Furthermore, various complications can arise during the course of the disease, which can affect multiple organ systems. These include sepsis, acute respiratory distress syndrome (ARDS)[1], and thromboembolic events such as pulmonary emboli[2], or myocardial infarction[3]. Since the beginning of the pandemic, there have been extensive efforts to identify causes and risk factors for severe COVID-19 progression. Several studies have demonstrated that higher age, male sex, and pre-existing conditions such as arterial hypertension (AHT) or diabetes mellitus (DM) may be risk factors for a more severe COVID-19 course and hospitalization[4,5].

One way to better understand COVID-19 disease is the use of quantitative omics approaches such as nuclear magnetic resonance (NMR) spectroscopy. With this method, a variety of metabolites and lipoproteins can be rapidly quantified in serum from numerous samples. Thus, a metabolic fingerprint of a disease can be identified, which can help to better understand pathogenesis, therapeutic effects, and to identify biomarkers for a more comprehensive diagnosis[6,7]. For COVID-19, several studies have investigated the metabolome and lipoproteome of patient sera both in the acute stage of infection and in the so-called post-acute COVID syndrome. In these studies, it is apparent that there are diverse changes in metabolites during COVID-19 infection. These include a decrease in various amino acids such as histidine, glutamine, lysine, methionine and tyrosine, whereas the amino acid phenylalanine showed increased levels in the studies in COVID-19 patients. Furthermore a general increase in ketone bodies was observed[8–10]. COVID-19 patients also showed signs of dyslipidemia with increased levels of very low density lipoprotein (VLDL) particles, triglycerides in total plasma and in lipoprotein subgroups. Further a decrease in high density (HDL) and low density lipoprotein (LDL) major and subgroups, which include phospholipids, cholesterol, and apolipoproteins A1 and A2 could be observed[8,11–14]. Also, the ABA1 ratio (apolipoprotein B100 divided by apolipoprotein A1) which is considered a cardiovascular risk marker[15], was increased in several COVID-19 cohorts[12,13]. In addition to the quantification of metabolites and lipoproteins in COVID-19 patients, it was possible to make statements about the outcome of patients[9]. Accordingly, in hospitalized patients with positive outcome, i.e., in Baranovicova et al. survivors without respiratory deterioration or mechanical ventilation, normalization of energy metabolism occurred more rapidly, as well as an increase in glutamine, than in patients with negative outcome (deterioration of respiratory situation with need for respiratory support or death). Furthermore, NMR technology has identified markers that may be indicative of a severe COVID-19 progression[10,16–18]. As such, above all, increased ketone bodies[16], phenylalanine[16,17], and pronounced dyslipidemia with increased ABA1 ratio, decreased HDL cholesterol[16], and decreased apolipoprotein A1[18] had been mentioned. These markers could help to predict response to therapy[16,19]. This was investigated in two studies with the IL-6 inhibitor tocilizumab. Under and after therapy, respectively, an approximation of the metabolic/lipoproteomic parameters of severe COVID-19 cases to the levels of milder cases was observed. Regarding the post-acute COVID syndrome (PACS), which can also present with a wide range of symptoms[20] and precise disease mechanisms are still unclear, NMR spectroscopy has shown that PACS is associated with delayed metabolic phenoreversion and that the metabolome and lipoproteome still differ from healthy controls one to three months after acute infection[21,22].

In addition to the metabolites and lipoprotein fractions discussed so far, N-acetylated- glycoproteins can be quantified by NMR as well. These include the Glycoprotein A (GlycA) signal, which is caused by various acute phase proteins such as haptoglobin, α1-acid glycoprotein, α1- antitrypsin, α1-antichymotrypsin and transferrin[23]. GlycA is combined with Glycoprotein B (GlycB), which represents N-acetylneuraminic acid[24], to form the Glyc signal and thus maps inflammatory processes[25]. Furthermore, Glyc is associated with increased cardiovascular risk[26,27]. Elevated Glyc levels were also found in COVID-19[14]. Another recent NMR parameter is the supramolecular phospholipids composite peak (SPC), which correlates mainly with apolipoproteins and HDL subfractions and is thus decreased in COVID-19 sufferers[28].

As mentioned earlier, cardiovascular disease and diabetes mellitus are among the risk factors for more severe COVID-19 progression[4]. These diseases also exhibit diverse alterations in the metabolome and lipoproteome. E.g. in patients with essential arterial hypertension, altered amino acid metabolism has been found, with decreased alanine and methionine levels, and increased arginine levels[29]. Lactate and pyruvate were also elevated in hypertensives in one study[30]. Furthermore, hypertensive patients show increased levels of cholesterol and triglycerides, as well as increased VLDL and LDL particles with reduced HDL and thus also an increased ABA1 ratio[31]. Elevated serum phenylalanine levels and monounsaturated fatty acids were markers of increased cardiovascular risk in another study[32]. Changes in amino acid metabolism are also found in type 2 diabetes mellitus (T2DM), indicating impaired energy metabolism[33]. In particular, impaired branched chained amino acid metabolism was conspicuous in this study in diabetic patients with late complications.

Cardiovascular disease has both a high prevalence and an impact on metabolites and lipoproteins concentrations in blood serum. Previous studies have taken this mostly not into account when investigating the metabolic phenotype of COVID-19 acute disease. Therefore, in this study, NMR data and laboratory parameters from 58 uninfected hypertensive individuals were compared with 216 samples of COVID-19-affected individuals also suffering from hypertension, and the influence of AHT on the previous findings on metabolic characteristics of COVID-19 was investigated. In our analyses, we demonstrate that the stratification for AHT results in a well-refined picture of the NMR-based metabolic phenotype of COVID-19. Furthermore, 509 samples of COVID-19-affected individuals with and without hypertension were compared to derive conclusions about risks that could drive a severe disease progression. Here, hypertensive patients with COVID-19 show metabolic cardiovascular risk factors and increased inflammatory parameters.

## Methods
An overview of all abbreviations is given in the Supplementary Tables 1 and 2.

**Patient cohort and sample collection.** The COVID-19 study population (enrollment from September 7th, 2020, to May 17th, 2021) consisted of patients monitored with the Coronataxi digital early warning (CDEW) system deployed in Rhein-Neckar County and Heidelberg, Germany. This is an outpatient care system consisting of remote digital monitoring via a mobile application (with symptom questionnaire and daily pulse oximetry), a medical doctor dashboard and medical care delivery to COVID-19 patients in home quarantine when indicated[34]. According to the

**Table 1 Characteristics of the COVID-19 cohort in total and sample collection.**

| Characteristics | Total | Female | Male |
|---|---|---|---|
| Sex, no. (%) | 329 | 174 (52.9) | 155 (47.1) |
| Age, median (IQR) | 54 (44–64) | 52 (41–63) | 58 (48–65) |
| BMI, median (IQR) | 27.9 (24.4–32.2) | 26.7 (22.6–32) | 29 (25.9–32.3) |
| Hospital treatment, no. (%) | 71 (21.6) | 25 (7.6) | 46 (14) |
| Death, no. (%) | 5 (1.5) | 3 (0.9) | 2 (0.6) |
| Arterial hypertension (%) | 134 (40.7) | 63 (19.1) | 71 (21.6) |
| Medication (antihypertensives) | | | |
| Beta-blockers (%) | 61 (18.5) | 31 (9.4) | 30 (9.1) |
| Calcium channel blockers (%) | 32 (9.7) | 14 (4.3) | 18 (5.5) |
| ACE- inhibitors (%) | 43 (13.1) | 17 (5.2) | 26 (7.9) |
| AT1-R-blockers (%) | 62 (18.8) | 34 (10.3) | 28 (8.5) |
| Sample collection period | | | |
| Timepoint of collection | 1 | 2 | 3 |
| Numbers of samples | 319 | 137 | 53 |
| Days after symptom onset, median (IQR) | 7 (5-9) | 10 (8-12) | 11 (10-14) |

The characteristics of the COVID-19 cohort are separated by sex. Besides information about anthropometrics, age and hospitalization, arterial hypertension (AHT) as thematized pre-existing condition in this work and the antihypertensive treatment are listed. Other pre-existing diseases and medication are listed in the Supplementary Tables 5 and 6. In some cases, multiple samples were collected per patient at a maximum of three time points. These are also given in the table together with the mean time interval after symptom onset in days.
*ACE* angiotensin-converting-enzyme, *AT1-R* angiotensin-1 receptor, *BMI* body mass index, *IQR* interquartile range.

**Table 2 Characteristics of the COVID-19 cohort, separated by regular blood pressure and AHT, and the AHT control cohort.**

| Characteristics | COVID-19 regular BP | COVID-19 AHT | AHT control cohort |
|---|---|---|---|
| Total | 195 | 134 | 58 |
| Sex, no. (%): | | | |
| Female | 111 (56.9) | 63 (47.0) | 29 (50) |
| Male | 84 (43.1) | 71 (53.0) | 29 (50) |
| Age, median (IQR) | 49 (37–60) | 62 (53–69) | 57 (52–61) |
| Hospital treatment, no. (%) | 38 (19.5) | 33 (24.6) | n.a. |
| CRP, median (IQR) | 18.1 (8.7–48.4) | 21.3 (8.3–57.8) | 0.4 (0.1–1.2) |

This table presents the relevant characteristics of the COVID-19 cohort with regular BP and AHT, as well as the AHT control cohort. Listed are the total patient numbers, sex, age, hospitalization, and the laboratory parameter CRP.
*AHT* arterial hypertension, *BP* blood pressure, *CRP* C-reactive protein, *IQR* interquartile range, *n.a.* not available.

**Table 3 Antihypertensive treatment of the AHT control cohort.**

| Antihypertensives | Total (%) | Female (%) | Male (%) |
|---|---|---|---|
| Beta-blockers (%) | 42 (72.4) | 21 (36.2) | 21 (36.2) |
| Calcium channel blockers (%) | 22 (37.9) | 11 (19) | 11 (19) |
| ACE- inhibitors (%) | 24 (41.4) | 14 (24.1) | 10 (17.2) |
| AT1-R-blockers (%) | 31 (53.4) | 15 (25.9) | 16 (27.6) |

This table shows the antihypertensive treatment with the four substance classes beta- blockers, calcium channel blockers, ACE-inhibitors, and AT1-R-blockers of the 58 patients in the AHT control cohort.
*ACE* angiotensin-converting- enzyme, *AT1-R* angiotensin-1 receptor.

enrollment period, the data were from the early phase of the pandemic, when the alpha virus or ancestral Wuhan strain was the predominant viral variant. Therefore, changes in findings due to viral evolution are possible. All patients were primarily affected. During the period between January and May 2021, six of the 329 COVID-19 patients included in this study (after quality control) were vaccinated with the vaccines from BionTech, AstraZeneca, and Moderna, respectively.

Based on the data obtained from the initial survey and all available self-reported data (e.g., self-reported dyspnea), a medical doctor at the university hospital decided whether a visit in home quarantine by a nurse was indicated and scheduled the visit for the following day.

Blood sample collection took place during home visits by nurses (the cohort and study therefore was called Coronataxi), which were carried out on medical indication, e.g., worsening of symptoms. Therefore, the blood samples were not taken a fastened state and there are various numbers of samples per study participant from zero to three. Also, the time intervals between blood collections varied. From a total of 459 participants in the Coronataxi study, a total of 543 samples from 348 COVID-19 patients was received and investigated by NMR. In interest to get the full statistical power, we took chance of using the NMR data of all samples, even there were several specimens of the same patient. This is a common approach in metabolomics studies. Blood was collected in Li-heparin or EDTA containing tubes. The specimens were then transported in isolated polystyrene boxes and centrifuged for serum preparation in the afternoon in the laboratory of Heidelberg University Hospital, Heidelberg, Germany. A frozen aliquot of the supernatant of these samples was transported to Tübingen, Germany on dry ice and stored at −80 °C until further processing and NMR analysis.

Besides common laboratory parameters (e.g., creatinine, C-reactive protein, white blood cell count and other) extensive metadata were provided by the University hospital Heidelberg, Heidelberg, Germany. This included information about anthropometrics, age, medication and pre- existing diseases. The metadata is listed in Table 1, and in the Supplementary Tables 3–6.

The cohort of the 58 arterial hypertension patients were provided by the interdisciplinary biomaterial- and databank of Würzburg University Hospital, Würzburg, Germany and were age and sex matched with the Heidelberger COVID-19 cohort. They were stored at −80 °C, until further processing. The metadata belonging to the serum samples was provided by the institute for clinical epidemiology and biometry Würzburg, Würzburg, Germany. Study enrollment of participants and collection of samples took place between July 2019 and February 2022. The pre-disease AHT was recorded as part of the patient's medical history. Further metadata contained information about age, sex, medication, diabetes mellitus as co- disease and common laboratory parameters, including renal values (creatinine, blood urea nitrogen, GFR), small blood count, inflammatory values (e.g., CRP), and isolated liver values. The relevant metadata for this work is listed in Table 2. An overview about the antihypertensive treatment is given in Table 3.

For improved statistical analysis, further NMR data (metabolites and lipoproteins) of an age and sex matched healthy control cohort (HC) of 305 subjects were provided by Bruker BioSpin GmbH, Ettlingen, Germany. These data did not contain any clinical laboratory parameters. An overview of all cohorts is given in Fig. 1.

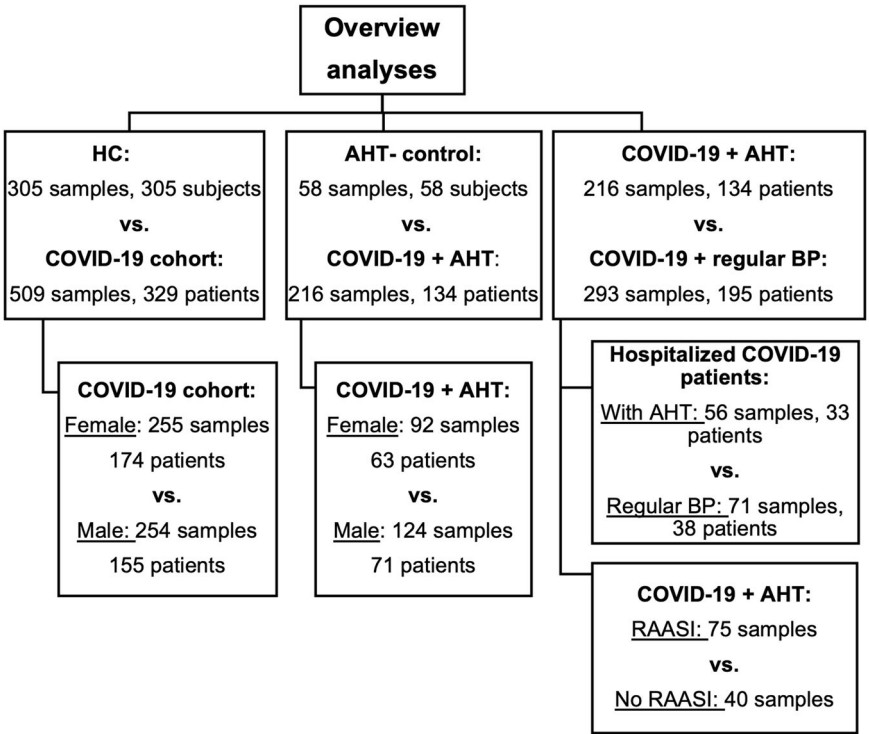

**Fig. 1 Overview of the cohorts and sample numbers.** Presented are patient and sample numbers, which were used for the various analyses. HC healthy control, AHT arterial hypertension, BP blood pressure, RAASI renin-angiotensin-aldosterone-system inhibitors.

**Sample preparation and NMR measurement**. The raw NMR spectra were acquired using Bruker's body fluids NMR methods package [https://www.bruker.com/de/products-and-solutions/mr/nmr-clinical-research-solutions/b-i-methods.html]. The sample preparation was performed following standard operating procedures (SOPs) to perform reproducible results. Briefly, blood serum samples were thawed to room temperature. Then 400 µl of commercially available plasma/serum buffer (Bruker BioSpin GmbH, Ettlingen, Germany) were pipetted into a 1,5 ml sterile Eppendorf tube. Next, an aliquot of 400 µl serum was added to the buffer. After gently shaken, an aliquot of 600 µl of the mixture was transferred into a 5 mm Bruker SampleJet NMR tube. Since only 350 µl was available for the cohort with the 58 hypertensive patients, 320 µl serum aliquot was added to 320 µl Bruker Plasma buffer (resulting in the same 1:1 mixing ratio) and an aliquot of 600 µl was used for measurement.

NMR spectroscopy was performed with a Bruker BioSpin IVDr Avance III HD 600 MHz system. It uses a temperature-controlled autosampler SampleJet™ for gentle handling during the measurements. 1D $^1$H-NMR spectra of the NOESY experiment (Nuclear Overhauser Spectroscopy) at a temperature of 310 K were acquired. NOESYs were analyzed using Bruker's software TopSpin (Version 3.6.2) and automatically processed. For the highest standardization and amount of reproducibility the following NMR-based commercial IVDr (in vitro diagnostics research) SOPs were used to collect the data sets for quality control, metabolites, lipoproteins, and inflammatory markers: B.I.BioBankQC$^{TM}$ in Plasma/Serum (quality control for Biobank blood bamples), B.I.Quant-PS$^{TM}$ (quantification of up to 41 metabolites in serum in mmol/L), B.I.LISA$^{TM}$ (lipoprotein analysis for 112 lipoprotein variables in serum), and B.I. PACS™ (quantification of two metabolites, nine lipoprotein variables, SPC and inflammatory markers e.g., GlycA/GlycB and Glyc/SPC ratio which provide information related to various complications in the context of post-acute COVID-19 syndrome).

Lipoprotein concentrations provided by the B.I. LISA™ package are divided in various subfractions broken down by triglycerides (TG), cholesterol (CH), apolipoproteins A1/-A2, apolipoprotein B100 bearing particles (AB), and total particle numbers (PN). These subfractions exist in the individual lipoprotein classes, very-low-density (VLDL), intermediate density (IDL), low-density (LDL), and high-density lipoproteins (HDL). In addition, the above mentioned subfractions (TG, CH, A1, A2, AB, PN) are also available in total plasma concentrations (TP). Phospholipid (PL) and free cholesterol (FC) fractions are also provided within the lipoprotein panel. Furthermore, the lipoproteins are subdivided according to size and thus increasing density. VLDL can be subdivided by NMR into five sizes, LDL into six, and HDL into four sizes. An increasing number is associated with smaller size or higher density. The extended names for the comprehensive lipoprotein panel are represented in the Supplementary Data 1.

Glycoproteins (Glyc A, Glyc B, Glyc) and the supramolecular phospholipid composite (SPC) data from the B.I. PACS™ module were available for the COVID-19 cohort and the AHT control cohort, but not for the HC from Bruker.

**Quality control, statistics, and data illustration**. To increase the quality of the NMR data set, the above-mentioned quality control reports of the individual samples were checked (B.I. BiobankQC$^{TM}$) and a total of 26 samples were sorted out based on these, mainly due to a linewidth of >2.3 Hz. This resulted in a total 509 samples from 329 COVID-19 patients and 58 samples from 58 hypertension patients were investigated.

The software package MetaboAnalyst 5.0 was used for statistical analysis of the obtained quantitative NMR reports. To change the data as little as possible, no normalization or scaling has been applied as NMR measurements are highly reproducible and results are based on absolute quantification. Only log transformation was performed to correct for heteroskedasticity. Features with >50% missing values have been removed. The other

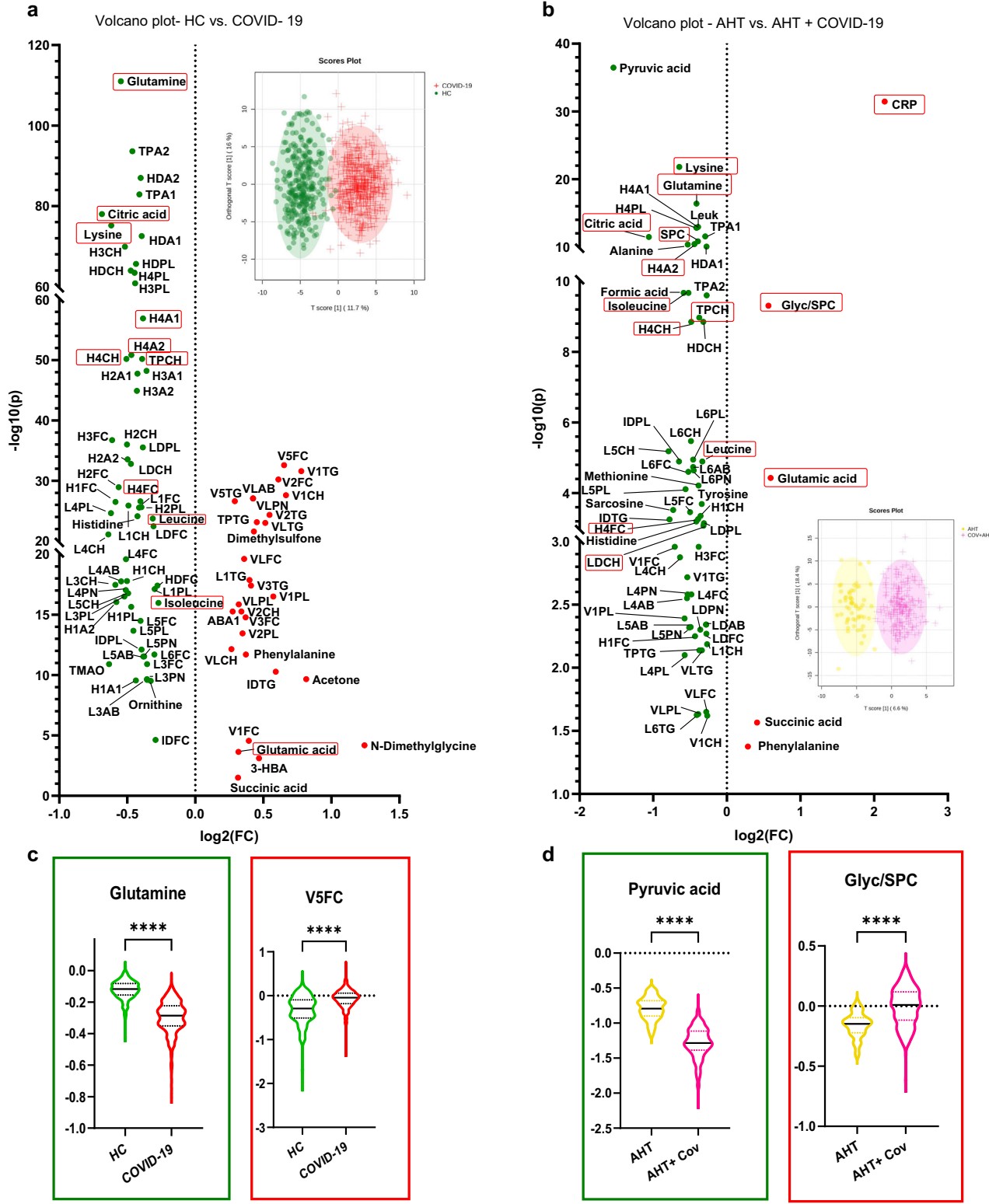

missing values were estimated with k-nearest neighbors (KNN) based on similar features. As univariate test method, volcano plots, which corresponds to a combination of unpaired *t*-test and fold change (FC) analysis, were used to determine significant alterations of metabolites and lipoproteins. Thereby a *p*-value threshold of <0.05 and FC threshold of 1.2 were set. As correction for multiple comparisons, false discovery rate (FDR) was used with the Benjamini-Hochberg procedure. FDR was performed for all statistical analyses with a threshold of FDR < 0.01. This is valid

for all data shown in Figs. 2–5 and in the Supplementary Data and Information. For multivariate statistics, unsupervised principal component analysis (PCA) was performed to get an overview of the clustering and to identify outliers. Features such as the 2D scores plots, loadings plots, and biplots were used for this purpose. Plots of multivariate statistics were used from MetaboAnalyst 5.0. Violin plots and volcano plots included in this article were created with GraphPad Prism 9. The flowchart (Fig. 1) was created with Microsoft PowerPoint 2019.

**Fig. 2 Univariate comparisons by means of volcano plots showing the differences of the metabolome and lipoproteome in COVID-19 + AHT patients compared to the COVID-19 cohort in total.** FC > 1.2, $p < 0.05$, FDR < 0.01: In the volcano plots (**a**) and (**b**), the red points on the right side of each plot are significantly increased in the COVID-19 cohorts, respectively, while the features marked with green points are significantly decreased in COVID-19 affected. Our analyses revealed metabolites and lipoproteins that were independent from sex and prior AHT and thus could be hypothesized as COVID-19 typical metabolic alterations in our cohort. These metabolites and lipoproteins are outlined in the red boxes. The plots with the orthogonal partial least squares discriminant analysis (OPLS-DA) additionally show to what extent the cohorts can be distinguished from each other. **a** shows the comparison between the COVID- 19 cohort in total ($n = 509$) with the HC ($n = 305$). **b** shows the comparison between hypertensive COVID-19 patients ($n = 216$) and uninfected hypertensive individuals ($n = 58$). In (**c**) and (**d**) violin plots with max. to min., whisker, and median, show the decreased (framed in green) respectively increased (framed in red) NMR parameters with the highest significance for each comparison (note here: CRP is not a NMR parameter but a laboratory value). The y-axis shows the normalized concentration. 3-HBA 3-hydroxybutyric acid, AHT arterial hypertension, Cov COVID-19, CRP C-reactive protein, HC healthy control, Leuk leukocytes count.

**Study approval**. This was a prospective non-interventional study conducted at the University of Heidelberg. Patients were eligible if they for were diagnosed with SARS-CoV-2 infection and if they had consented to study participation. Written informed consent according to the Declaration of Helsinki was obtained for all patients. The local ethics committee of the University of Heidelberg, Heidelberg, Germany had approved data collection of the COVID-19 cohort and analysis of the obtained samples (reference number: S-324/2020). The patient recruitment for the AHT control cohort took place in Tübingen, Germany with the ethics vote with the number 141/2018BO2, dated May 4, 2022. Since the data were kept in Würzburg, Germany, there is a second ethics vote with the number 52/18. We confirm that all relevant ethical regulations were followed.

**Reporting summary**. Further information on research design is available in the Nature Portfolio Reporting Summary linked to this article.

## Results
**Cohort composition**. The main characteristics of the Coronataxi cohort are shown in Table 1. As this cohort was divided in a normotensive and hypertensive COVID-19 patient group, some characteristics for these are listed separately in Table 2, along with the AHT control cohort. All hypertensive COVID-19 patients were diagnosed with chronic AHT before the COVID-19 pandemic. There was no data of the exact duration and severity of illness. 120 of the 134 hypertensive COVID-19 patients were under antihypertensive treatment at the time of the study. Table 3 gives an overview of the antihypertensive treatment in the AHT control cohort.

**Overview of the performed analyses and results**. In this work, NMR data from a healthy cohort ($n = 305$) were compared with NMR data of the COVID-19 cohort in total ($n = 509$). Further, NMR data and laboratory parameters of COVID-19-affected individuals with AHT ($n = 216$) were compared with an AHT control cohort ($n = 58$) to investigate the impact of AHT on previous findings on metabolomic, lipoproteomic and inflammatory changes in COVID-19 patients. These comparisons were additionally analyzed separately by sex. NMR data and laboratory parameters were also analyzed from COVID-19 patients with AHT ($n = 216$) and without ($n = 293$). The same was performed with samples of hospitalized COVID-19 patients, of which 33 individuals (sample $n = 56$) suffered from AHT, while 38 patients (sample $n = 71$) had a regular blood pressure. Last, AHT patients with different antihypertensive treatment were compared. An overview of all performed analyses is given in Fig. 1.

In Table 4 an overview of the results of the individual univariate analyses is provided. Regardless of sex and prior AHT (columns 1, 4, 5), all COVID-19 patients showed an increase in

CRP, decreases in alanine, glutamine, lysine, pyruvate, and in the lipoprotein subgroups H4A1, H4CH, and H4PL. Other common features not included in this presentation, due to a smaller AUC but still showing significant changes in univariate statistics, were an increase in Glyc/SPC ratio, decreased levels of citric acid, isoleucine, and leucine, as well as HDL- subgroups H4A2 and H4FC, total cholesterol (TPCH), total apolipoproteins (TPA1, TPA2), and SPC. Interestingly, COVID-19 groups had decreased leukocyte counts in contrast to the control group. There were differences of laboratory parameters in COVID-19 affected depending on sex (column 2). This included higher levels of creatinine, ferritin, gamma-glutamyl transferase (GGT) and blood urea nitrogen (BUN) in males. It was also noticeable that men had higher inflammatory parameters in the form of Glyc/SPC. The branched chained amino acids (BCAA) leucine and valine were increased in men. HDL subclasses (HDL-1, -2, -3) were decreased in males. When COVID-19 + AHT affected women and men were compared (column 3), there were some changes in the sex comparison. In this subgroup, the BCAAs leucine and isoleucine were increased in male patients, but the changes in lipoproteins now concerned LDL subgroups (mainly the larger particles LDL-1 and -3), which showed lower levels in men than in women. When COVID-19 patients with normal blood pressure and AHT were compared (column 6), metabolic markers associated with AHT were conspicuous in the hypertensive COVID-19 group. These included elevated creatinine and BUN, a triglyceride-rich lipoprotein profile (IDTG, TPTG, VLTG, VLDL-1, -2), and the intestinal metabolite trimethylamine-N-oxide (TMAO). The last mentioned was found to be elevated only in the COVID-19 + AHT group. Hospitalized hypertensive COVID-19 patients (column 7), also showed a triglyceride-rich lipoprotein profile compared with hospitalized normotensive COVID-19 patients. This was characterized by increased total and VLDL triglycerides (TPTG, VLTG), VLDL particles (VLAB, VLDL-1, VLDL-2, VLDL-3), and, in contrast to the whole COVID-19 + AHT group, also by an increase in HDL triglycerides (HDTG, H1TG, H2TG, H3TG). All group comparisons discussed here, as well as additionally the comparison of the whole COVID-19 cohort with Bruker's HC and the comparison of different antihypertensive treatment groups, will be discussed in detail in the next sections.

**Differences in the metabolomic phenotypic when comparing total COVID-19 patient cohort with sub-cohort of patients with hypertension**. For comparison with previously reported findings of NMR-based alterations of metabolites and lipoproteins, a volcano plot analysis was performed comparing the 305 samples of the HC with the entire Coronataxi COVID-19 cohort, i.e., 509 samples. All metabolic changes named below therefore apply to the comparison of the entire COVID-19 cohort with Bruker's HC and can be seen in the volcano plot in Fig. 2a. In this plot, 60 metabolites and lipoproteins were decreased more than

**Table 4 Overview of results of the individual group comparisons in the univariate statistics.**

| Parameter | 1. COVID-19 + AHT vs. AHT | 2. COVID-19: Male vs. Female | 3. COVID-19 + AHT: Male vs. Female | 4. Female: COVID-19 + AHT vs. AHT | 5. Male: COVID-19 + AHT vs. AHT | 6. COVID-19: AHT vs. Regular BP | 7. Hospitalized COVID-19: AHT vs. Regular BP |
|---|---|---|---|---|---|---|---|
| Alanine | 0.75↓**** | | | 0.73↓**** | 0.76↓**** | | |
| BUN | | 1.45↑**** | 1.57↑**** | | | 1.39↑**** | |
| Citric acid | 0.69↓**** | | | | | | |
| Creatine | | | | | | | 0.46↓* |
| Creatinine | | 1.44↑**** | 1.43↑**** | | | 1.22↑**** | 1.35↑*** |
| Creatinine.lab | | 1.47↑**** | 1.61↑**** | | | 1.24↑**** | 1.51↑*** |
| CRP | 4.41↑**** | | | 4.37↑**** | 4.27↑**** | | |
| DMSO | | | | | 0.52↓** | | |
| Ferritin | | 3.11↑**** | | | | | |
| Formic acid | 0.48↓**** | | | | 0.44↓**** | | |
| GGT | | 2.16↑**** | | | | | |
| Glutamine | 0.75↓**** | | | 0.78↓**** | 0.73↓**** | | |
| Glyc/SPC | | 1.26↑**** | | | 1.46↑*** | | |
| H1A1 | | 0.73↓**** | | | | | |
| H1A2 | | 0.7↓**** | | | | | |
| H1CH | | 0.74↓**** | | | | | |
| H1FC | | 0.74↓**** | | | | | |
| H1PL | | 0.73↓**** | | | | | |
| H1TG | | | | | | | 1.30↑* |
| H2CH | | 0.83↓**** | | | | | |
| H2TG | | | | | | | 1.26↑* |
| H3FC | | 0.8↓**** | | | | | |
| H3TG | | | | | | | 1.23↑* |
| H4A1 | 0.76↓**** | | | 0.77↓**** | 0.76↓**** | | |
| H4A2 | 0.74↓**** | | | | 0.72↓**** | | |
| H4CH | 0.71↓**** | | | 0.72↓**** | 0.71↓**** | | |
| H4PL | 0.75↓**** | | | 0.76↓**** | 0.75↓**** | | |
| HDA1 | 0.83↓**** | | | 0.80↓**** | | | |
| HDA2 | | | | 0.83↓**** | | | |
| HDCH | | | | 0.77↓**** | | | |
| HDTG | | | | | | | 1.23↑* |
| IDTG | | | | | 1.43↑*** | | |
| Isoleucine | | | 1.28↑*** | | 0.66↓**** | | |
| L1AB | | | 0.82↓**** | | | | |
| L1CH | | | 0.8↓**** | | | | |
| L1FC | | | 0.82↓*** | | | | |
| L1PL | | | 0.81↓**** | | | | |
| L1PN | | | 0.82↓**** | | | | |
| L3AB | | | 0.78↓** | | | | |
| L3CH | | | 0.76↓*** | | | 0.79↓*** | |
| L3FC | | | 0.8↓** | | | | |
| L3PL | | | 0.78↓** | | | 0.82↓*** | |
| L3PN | | | 0.79↓** | | | | |
| Leucine | | 1.23↑**** | 1.35↑**** | | | | |
| Leuk | 0.70↓**** | | | 0.67↓**** | 0.71↓*** | | |
| Lysine | 0.64↓**** | | | 0.63↓**** | 0.64↓**** | | |
| Methionine | | | | | 0.72↓**** | | |
| Pyruvic acid | 0.34↓**** | | | 0.34↓**** | 0.34↓**** | | |
| SPC | 0.76↓**** | | | 0.73↓**** | | | |
| TPA1 | 0.82↓**** | | | 0.79↓**** | | | |
| TPA2 | | | | 0.80↓**** | | | |
| TPCH | | | | | 0.73↓**** | | |
| TPTG | | | | | | 1.22↑**** | 1.31↑* |
| TMAO | | | | | | 1.25↑* | |
| V1CH | | | | | | 1.31↑*** | 1.48↑* |
| V1PL | | | | | | 1.34↑*** | 1.43↑* |
| V1TG | | | | | | 1.34↑*** | 1.45↑* |
| V2FC | | | | | | 1.23↑** | 1.43↑* |
| V2TG | | | | | | 1.23↑** | |
| V3FC | | | | | | | 1.45↑* |
| Valine | | 1.2↑**** | | | | | |
| VLAB | | | | | | | 1.23↑* |
| VLTG | | | | | | 1.26↑*** | 1.34↑* |

This table gives an overview of the performed univariate analyses. Here the Biomarker analysis from Metaboanalyst 5.0 was used to elaborate the most prominent alterations of the respective comparisons. From all significantly altered metabolites, lipoproteins, and laboratory parameters in the volcano plot analyses, those 15 with the highest area under the curve (AUC) were chosen. The number indicated is the fold change (FC). The arrow indicates whether a metabolite, lipoprotein or laboratory value was found to be decreased or increased in the group first mentionend, respectively. The p-value is indicated in form of asterisks (*$p < 0.05$,**$p < 0.01$,***$p < 0.001$,****$p < 0.0001$). Since many metabolites were found to be significant in individual comparisons, the 15 that also showed the highest area under curve (AUC) in the MetaboAnalyst 5.0 Biomarker analysis were selected for this presentation. The abbreviations for the lipoproteins are listed in the Supplementary Data 1.
*AHT* arterial hypertension, *BP* blood pressure, *BUN* blood urea nitrogen, *Creatinine.lab* creatinine value from laboratory report, *CRP* C-reactive protein, *GGT* gamma-glutamyl-transferase, *Glyc* glycoproteins, *Leuk* leukocytes count, *SPC* supramolecular phospholipid composite, *TMAO* trimethylamine-N-oxide.

1.2-fold, whereas 29 were increased. The metabolic changes largely coincided with the literature already cited in the introduction. A triglyceride-rich lipoprotein profile could be detected in the total COVID-19 cohort based on the elevated levels of total plasma triglyceride (TPTG), other triglyceride subgroups (IDTG, L1TG), all major VLDL groups (VLAB, VLPN, VLCH, VLFC, VLTG, and VLPL), and several VLDL subgroups (e.g., V1TG, V2CH, V3TG, V5FC, for example) in all sizes except VLDL-4. On the other hand, a reduction in total plasma apolipoproteins A1 and A2 and all major HDL- groups (HDA1, HDA2, HDCH,

HDFC, HDPL), except HDTG, was observed in the COVID-19 cohort. All four sizes were present, with HDL-3 and -4 as the most significantly altered subgroups, of which the apolipoprotein, phospholipid, and cholesterol subgroups were particularly represented. Furthermore, total plasma cholesterol (TPCH), several LDL major (LDCH, LDFC, LDPL), and subgroups, were shown to be decreased. This was mainly true for denser LDL subgroups (LDL-3, -4, -5, -6), as well as for the phospholipid and cholesterol subgroups (e.g., L4CH, L5PL, L6FC). For metabolites, we observed a decrease over 1.2-fold in several amino acids, including glutamine, lysine, histidine, leucine, and isoleucine. Another reduced metabolite was citric acid. In contrast, ketone bodies (3-hydroxybutyric acid, acetone), the amino acids phenylalanine and glutamic acid, and the metabolites succinic acid and dimethylglycine were increased in the COVID-19 cohort. For multivariate comparisons of the COVID-19 cohort in total with the HC, see Supplementary Fig. 1.

To see whether the results described above could be reproduced when stratifying for AHT, the COVID-19 cohort was divided into a hypertensive and a normotensive sub-cohort. The COVID-19 cohort with AHT was compared with an AHT control group without COVID-19 using another volcano plot (Fig. 2b). We could identify 60 metabolites, lipoproteins, and laboratory parameters which showed a decrease of more than 1.2-fold in the COVID-19 + AHT cohort (as in the previous comparison), while 5 showed an increase (versus 29 increased features in the comparison COVID-19 in total vs. HC). The main differences between those two comparisons were that we could not observe a triglyceride-rich lipoprotein profile in the COVID-19 + AHT cohort. Only the inflammatory parameters CRP and Glyc/SPC ratio, the amino acids glutamic acid and phenylalanine, and the citrate cycle metabolite succinic acid were elevated. The above-described increase in ketone bodies did not show up in the COVID-19 + AHT group. Furthermore, the significant reductions in HDL- fractions in COVID-19 + AHT patients were mainly concentrated on HDL-4 particles (H4A1, H4A2, H4CH, H4FC, H4PL), in contrast to the whole COVID-19 group. On the other hand, there were some features, which these two comparisons had in common, including decreases in amino acids lysine, glutamine, histidine, leucine, and isoleucine in the SARS-CoV-2 infected patients with and without AHT. In the COVID-19 + AHT cohort, additionally, the amino acids alanine, methionine, and tyrosine were decreased. Another common metabolite was citric acid, which showed decreased levels in COVID-19 patients in total and in COVID-19 + AHT. As in the whole COVID-19 cohort, also in the hypertensive group, a reduction of TPCH and several LDL- major- (LDAB, LDCH, LDFC, LDPL) and subgroups could be observed. Again, mostly denser particles (LDL-4, -5, -6) and phospholipid and cholesterol fractions of LDL were present. Interestingly, a few major VLDL groups (VLFC, VLPL, VLTG) and isolated VLDL-1 particles were also reduced in the COVID-19 + AHT cohort. For multivariate comparisons of the COVID-19 + AHT cohort with the AHT control cohort, see Supplementary Fig. 2.

**Sex differences in COVID-19 patients with and without AHT and metabolic characteristics in our COVID-19 cohort.** One aim was to test whether there were metabolic sex differences in our COVID-19 cohort and to eliminate the possible confounder sex. For this purpose, the total COVID-19 cohort was divided into male and female and analyzed using univariate statistical operations. The same was done for the COVID-19 + AHT cohort. The results are shown in the two volcano plots in Supplementary Fig. 3. Multivariate analyses for the sex comparisons are shown in Supplementary Figs. 4 and 5.

In the sex comparison of the total cohort 26 features were significantly increased in male COVID-19 patients, whereas 19 were decreased. Overall, some sex-specific laboratory parameters were altered, such as creatinine, blood urea nitrogen (BUN), ferritin, and the liver values GGT and GOT, which showed higher values in men than in women. Furthermore, males with COVID-19 showed higher levels of Glyc/SPC ratio, BCAAs, ketone bodies, and various VLDL- subgroups, with triglyceride, phospholipid, and cholesterol fractions. In comparison, women showed higher values in HDL-subgroups. HDL-4 did not stand out in the sex comparison. Furthermore, LDL particles were more abundant in female COVID-19 patients.

When the COVID-19 + AHT cohort was analyzed, there were fewer metabolic differences between the sexes. 10 features in males (vs. 26 in the male COVID-19 group in total) were significantly increased compared to females; 15 features (vs. 19 in the male COVID-19 cohort in total) were decreased. The triglyceride-rich lipoprotein profile of men now was less pronounced. Ketone bodies no longer stood out as significant, nor did the Glyc/SPC ratio. Men showed higher levels of the BCAAs leucine and isoleucine, as well as citric acid, and dimethylsulfone. Women continued to show higher values in the LDL fractions, but not in the HDL particles.

The comparisons of COVID-19 + AHT with an AHT control group, as well as the sex comparisons, identified characteristics that are independent of sex and prior AHT disease and thus typical of COVID-19 infection in our cohort. In the volcano plots in Fig. 3, male and female hypertensive patients with and without COVID-19 were compared. There were 29 samples from 29 female hypertensive patients without COVID-19 and 92 samples from 63 female hypertensive patients with COVID-19 and AHT (Fig. 3a). In Fig. 3b, 29 samples from 29 hypertensive male patients without and 124 samples from 71 male patients with COVID-19 and AHT were compared. In these analyses, an increase in Glyc/SPC, glutamic acid, and CRP could be observed. On the other hand, there were decreases in amino acids leucine, isoleucine, glutamine, lysine, citric acid, HDL-4 fractions (H4A1, H4A2, H4CH, H4FC, H4PL), total cholesterol (TPCH), and SPC. These mentioned metabolites and lipoproteins were also apparent in the previous univariate statistics (Fig. 2). Thus, they can be assumed to be independent of sex and AHT and can be hypothesized to represent COVID-19 typical metabolic changes in our cohort. For better illustration, these markers are framed in red in Figs. 2 and 3. Multivariate analyses for the sex comparisons, respectively, can be seen in Supplementary Figs. 6 and 7.

**Hypertensive COVID-19 patients show NMR-based metabolomic and lipoproteomic characteristics of an increased cardiovascular risk.** The aim of the next analysis was to compare hypertensive and normotensive COVID-19 patients to investigate if there are metabolic alterations between those two groups, which could indicate a higher risk for a severe COVID-19 disease course. 216 samples from 134 hypertensive COVID-19 patients were compared with 293 samples from 195 normotensive COVID-19 patients. A volcano plot (Fig. 4) was performed, where 15 metabolites, lipoproteins, and laboratory parameters were increased more than 1.2-fold. Two lipoprotein subclasses (L3CH, L3PL) and one metabolite (dimethylsulfone) were reduced. The highest significant alterations were seen in the elevated renal values in the hypertensive COVID-19 group. BUN and creatinine.lab corresponded to standard laboratory parameters, whereas creatinine was part of the NMR-based B.I. QUANT-PS report. In addition, significant increases in triglyceride-rich VLDL particles were observed, especially in the

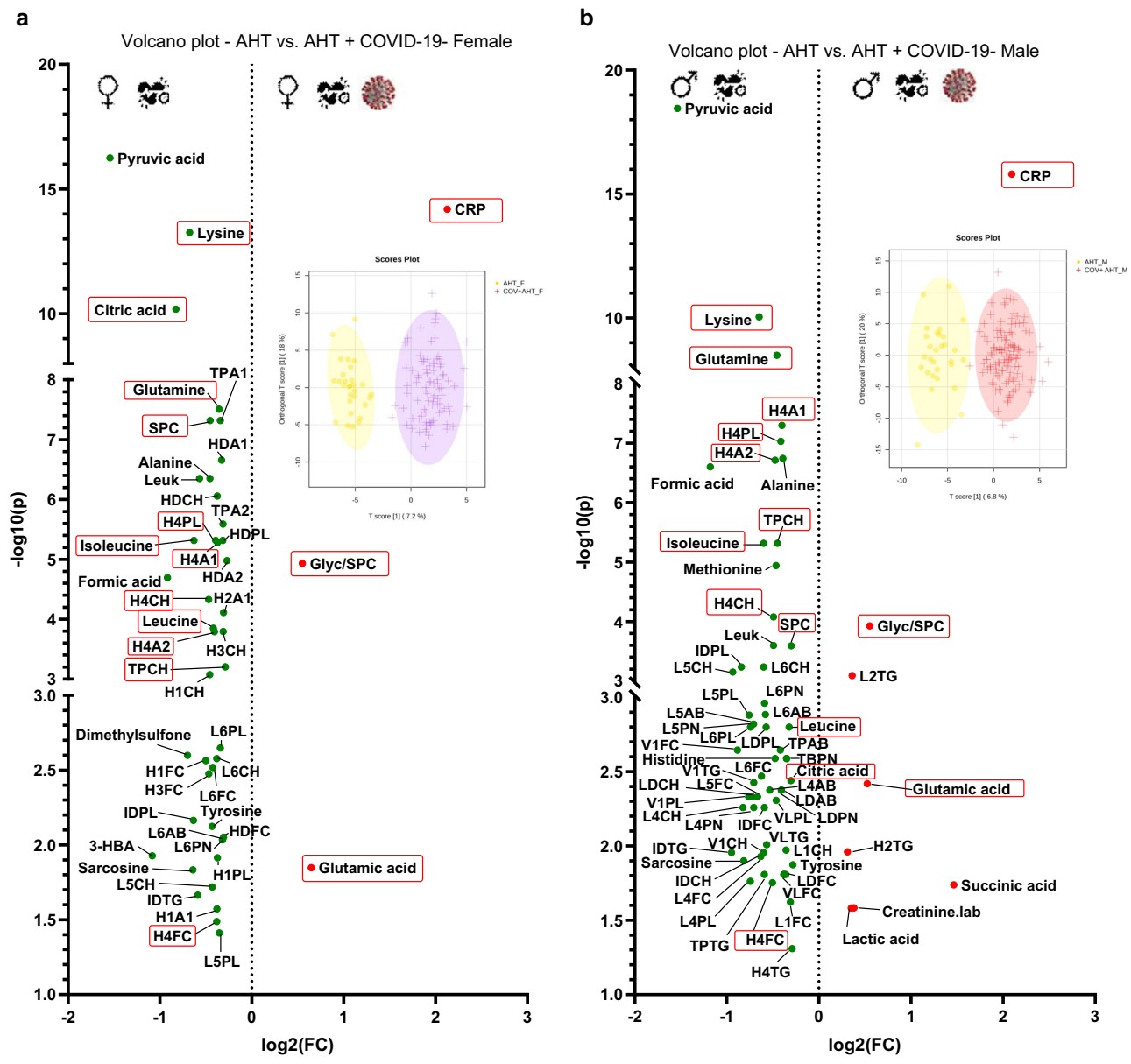

**Fig. 3 Univariate comparison of AHT cohorts with and without COVID-19, separated by sex, respectively.** FC > 1.2, *p* < 0.05, FDR < 0.01: In the volcano plots (**a**) and (**b**), the red points on the right side of each plot are significantly increased in the COVID-19 + AHT cohorts, respectively, while the features marked with green points are significantly decreased in COVID-19 + AHT affected. Our analyses revealed metabolites and lipoproteins that were independent from sex and prior AHT and thus could be hypothesized as COVID-19 typical metabolic alterations in our cohort. These metabolites and lipoproteins are outlined in the red boxes. The plots with the orthogonal partial least squares discriminant analysis (OPLS-DA) additionally show to what extent the cohorts can be distinguished from each other. Plot (**a**) shows the comparison of AHT (*n* = 29) vs. COVID-19 + AHT (*n* = 92) for females, plot (**b**) shows this comparison for male sex (AHT male: *n* = 29; COVID-19 + AHT male: *n* = 124). 3-HBA 3-hydroxybutyric acid, AHT arterial hypertension, Creatinine.lab creatinine from laboratory report, CRP C-reactive protein.

larger and less dense subgroups (VLDL-1, -2, -3), as well as in the triglyceride subfractions (VLTG, V1TG, V2TG). Total plasma triglyceride (TPTG) and IDL triglyceride fraction (IDTG) also showed elevated levels. Regarding metabolites, the citrate cycle metabolite succinic acid was increased, as in other analysis. The metabolite trimethylamine-N- oxide (TMAO), which is produced by intestinal bacteria and is considered a cardiovascular risk marker, only stood out in this analysis. For multivariate comparisons of those groups, see Supplementary Fig. 8.

Similar metabolic alterations were observed in hospitalized COVID-19 + AHT patients (Supplementary Fig. 9). This group also showed elevated renal values (BUN, creatinine) and a triglyceride-rich lipoprotein profile compared with normotensive

hospitalized COVID-19 patients. However, the lipoprotein subclass distribution differed from the COVID-19 + AHT group in total in that HDL triglycerides were elevated in the hospitalized COVID-19 + AHT group. These subgroups were not conspicuous in the previous comparison (Fig. 4). Supplementary Fig. 10 shows the multivariate analysis.

Of the 134 hypertensive COVID-19 affected patients, 120 were treated with antihypertensives. 48 of these were treated with one medication, 72 with two or more. An overview of the medications of the hypertensive patients was given in Table 1. Antihypertensives indicated were angiotensin-converting-enzyme inhibitors (ACEI), angiotensin-1 receptor blockers (AT1RB), beta-blockers (BB), and calcium channel blockers (CCB). To determine any

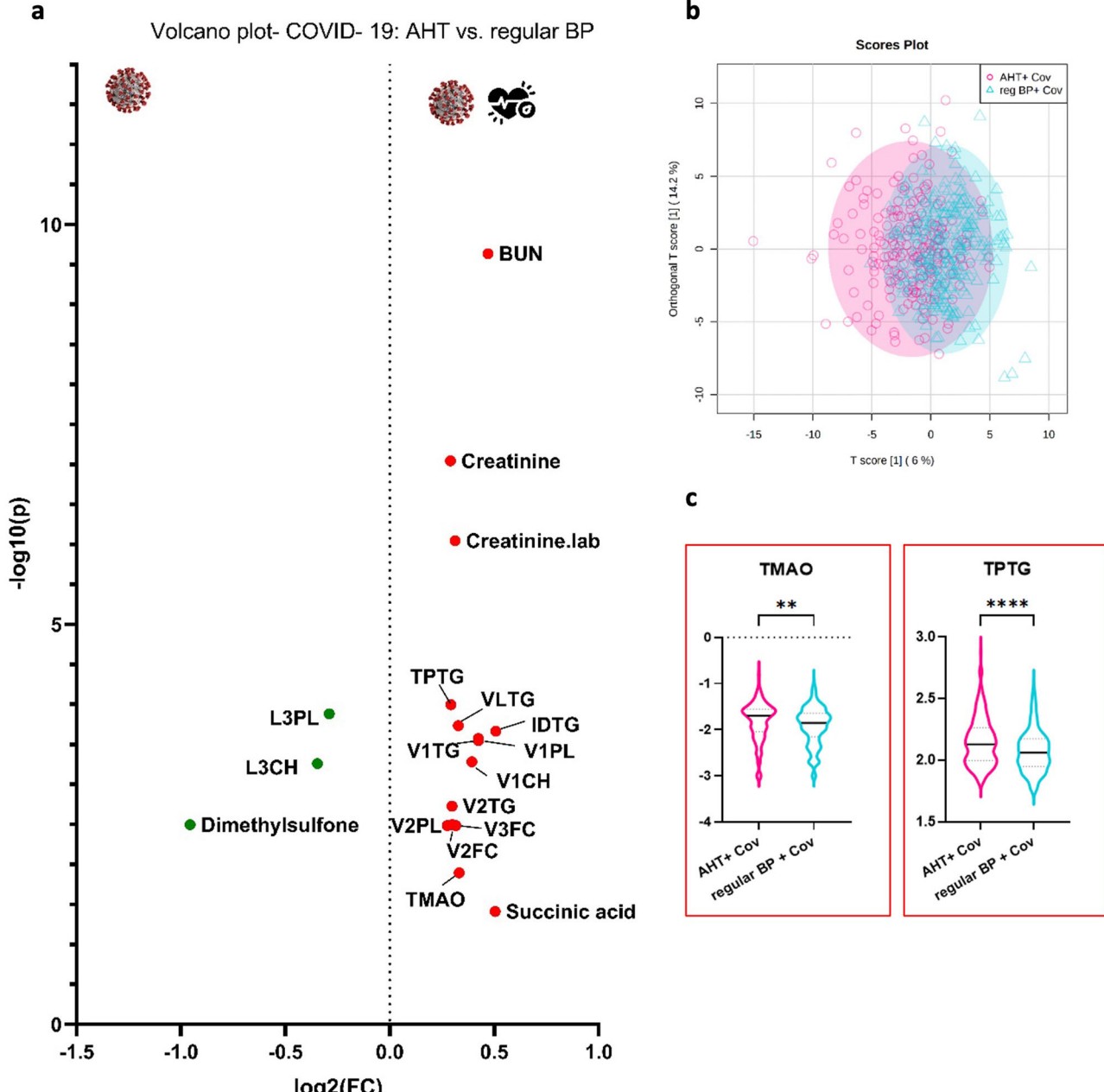

**Fig. 4 Metabolomic differences in normotensive and hypertensive COVID-19 patients in univariate and multivariate statistics.** In (**a**) univariate analysis by means of volcano plot showing significant differences in normotensive and hypertensive COVID-19 patients (FC > 1.2, $p < 0.05$, FDR < 0.01): The features marked with red points on the right side are increased in the COVID-19 + AHT cohort ($n = 216$), while the green ones are decreased in COVID-19 + AHT compared to normotensive COVID-19 patients ($n = 293$). **b** Multivariate analysis by means of orthogonal partial least squares discriminant analysis (OPLS-DA) showing the distribution of both groups. **c** Violin plots with max. to min., whisker, and median, showing significant alterations of the cardiovascular risk marker trimethylamine-N-oxide (TMAO) and TPTG, whose significance will be examined in more detail in the discussion. The y-axis shows the normalized concentration. AHT arterial hypertension, BP blood pressure, BUN blood urea nitrogen, Cov COVID-19, Creatinine.lab creatinine from laboratory report.

medication differences in metabolites and lipoproteins, the COVID-19 + AHT cohort was divided into subgroups according to their antihypertensive treatment. Patients treated with ACEI and/or AT1RB alone were grouped as the renin-angiotensin-aldosterone system inhibitor (RAASI) group, which included 75 samples. Hypertensive patients treated with BB and/or CCB formed the no RAASI group, which included 40 samples. The majority in this group were treated with BB (29 samples). 75 samples were from patients treated with different combinations of ACEI/AT1RB and BB and/or CCB. The 26 samples from

patients who were not taking antihypertensives according to metadata were excluded from this analysis. The groups were compared with univariate operations. Here, the RAASI- group showed improved renal values compared with the no RAASI group, in terms of decreased creatinine levels and higher glomerular filtration rate (GFR). This was not observed when patients were treated with a combination of RAASI and BB or CCB. In general, no significant alterations could be seen in the comparison of a combination of RAASI with BB/CCB versus the no RAASI group. Interestingly, increased VLDL particles

(VLPN), consisting of the denser subgroups VLDL-4, were observed in the RAASI group. All significant changes are shown in the volcano plot in Supplementary Fig. 11. Multivariate analysis of this comparison is presented in Supplementary Fig. 12.

**Hypertensive COVID-19 patients show higher inflammatory NMR parameters**. As a further univariate test method, an unpaired *t*-test was performed with the cohorts AHT versus COVID-19 + AHT. Another unpaired t-test was performed for the comparison COVID-19 with AHT versus regular BP. The *p*-value threshold was set at 0.05 (FDR < 0.01). These *t*-tests were performed for the already mentioned typical altered metabolites (glutamine, isoleucine, leucine, lysine, citric acid) and lipoproteins (HDL-4, TPCH, LDCH, SPC) in COVID-19 patients. The results are shown using violin plots in Supplementary Fig. 13. It was additionally observed that hypertensive COVID-19 patients showed higher inflammation levels than normotensives. This was true for the NMR parameters Glyc, as well as the Glyc/SPC ratio, with the hypertensive COVID-19 group showing significantly higher values with *p*-values of $p = 0.0014$ for Glyc and $p = 0.0058$ for Glyc/SPC ratio. SPC was decreased in normotensive COVID-19 patients compared to the cohort without COVID-19 and therefore did not stand out as significant in this comparison, same for the laboratory value CRP, which was increased in normotensive and hypertensive COVID-19 patients. Violin plots for the mentioned inflammatory parameters are shown in Fig. 5.

A summary of this work is presented in Fig. 6 and an exemplary assigned NMR spectrum for metabolites, glycoproteins and SPC is shown in Supplementary Fig. 14.

## Discussion

In this study, we analyzed a COVID-19 cohort with 509 serum samples from 329 patients, which were treated mainly in an outpatient care setting. In our analyses, we focused on 134 patients (216 samples), which were additionally affected by arterial hypertension. To the best of our knowledge, this is the first study to investigate the effect of AHT on COVID-19 in the context of NMR-based metabolomics. We could show that the metabolic changes of COVID-19 disease behave differently when pre-existing AHT is considered. Stratification for AHT has also an impact on differences detected between sexes. Furthermore, it became clear that hypertensive COVID-19 patients might at higher risk for complications, shown by certain changes in lipoproteins, metabolites, and inflammatory parameters.

The Coronataxi study took place in Germany, where, according to the Robert-Koch- Institute, AHT prevalence is about one-third in women and men, respectively. In subjects older than 65 years prevalence is even two third for both sexes. In our COVID-19 cohort AHT was also very present with 40.7%. Previously it has been shown that individuals with pre-existing cardiovascular disease have an altered metabolic profile, particularly with respect to lipoproteins. These include, among others, increased VLDL particles and decreased HDL levels[31,35]. Since hypertensive patients may thus have elevated VLDL levels, they obviously drop out in the univariate statistics when COVID-19 patients with hypertension are compared with controls with hypertension, leading to the hypothesis that COVID-19, at least in milder courses, does not cause an increase in VLDL and its subgroups, as well as in the ABA1 ratio. However, what does suggest dyslipidemia in COVID-19-affected individuals despite this are the decreased levels in the HDL fractions. In particular, the decreased levels in the dense HDL-4 particles are considered typical of COVID-19 in our cohort. HDL appears to correlate inversely with cardiovascular risk[36]. However, there are discrepancies in the extent to which the various HDL subgroups are

involved in this process[37]. Furthermore, since different methods are used to measure HDL subgroups, it is not always easy to compare the literature in this regard. Small HDL particles, such as HDL-4 (corresponding to HDL-3 in ultracentrifugation[38]), have been shown in some studies to be risk factors for cardiovascular disease[39]. On the other hand, the protective and antioxidant effects on the endothelium[40] and the fibrinolytic properties of denser HDL particles had been described[41]. Apolipoproteins A1 and A2 each play a major role, as well as inverse correlating markers for cardiovascular risk[42]. Adjacent to cardiovascular disease, HDL has a more clear relationship with the immune system[43,44] and changes in its composition and function during an acute phase reaction[45]. This is characterized by a decrease in apolipoprotein A1 and HDL cholesterol, which was evident in our COVID-19 cohort where we identified a decrease in total (TPA1 and TPCH, respectively) and HDL groups (HDA1 and H4A1, and HDCH and H4CH, respectively). Thus, it can be inferred that the acute-phase response induced by COVID-19 may cause dysfunctionality of HDL particles, leading to loss of atheroprotective properties[46]. In addition to the decrease in HDL-particles, several LDL- fractions also showed a decrease. This is consistent with other studies[8,14]. Furthermore, a correlation between increasing disease severity and a decrease in LDL cholesterol, as well as an inverse correlation between LDL cholesterol and CRP could be observed[47,48]. Possible reasons for this include impaired liver function, altered lipid metabolism under the influence of cytokines, and the breakdown of lipids by free radicals[47].

Regarding metabolites, COVID-19 characteristics include decreases in amino acids glutamine, isoleucine, leucine, and lysine. This indicates an altered energy metabolism in the course of hypoxemia[49]. Decreased glutamine correlated inversely with CRP, lactate dehydrogenase (LDH), and partial pressure of oxygen in one study[49]. In addition, glutamine fulfills important functions in the body, e.g., as a reactant for glutathione biosynthesis or as an energy supplier for immune cells[50]. The reduction in the other amino acids mentioned could indicate increased amino acid catabolism, possibly as a result of temporary fasting during acute infection, as of our COVID-19 cohort, 22% reported loss of appetite and 9% nausea. Another metabolite that is decreased in COVID-19 is citric acid. This Krebs cycle metabolite also correlated inversely with CRP and LDH in one study[51].

Comparing the AHT-control cohort with the COVID-19 + AHT cohort, the acute phase protein CRP showed an increase, which is not surprising, as well as the NMR-based inflammatory parameter Glyc/SPC. This ratio was also elevated in another study of COVID-19 patients[28]. Glycoproteins (Glyc) represent the acute phase reaction[23]. Supramolecular phospholipid composite signal (SPC) reproduces lipoproteins, especially apolipoproteins and HDL[28]. Another prominent parameter in the COVID-19 cohort was phenylalanine. It has already been mentioned that this amino acid is elevated in COVID-19 patients and also correlates with a more severe progression[17]. Even before the COVID-19 pandemic, phenylalanine was shown to be elevated in severe infections and associated with increased mortality, intensive care unit (ICU) treatment, and CRP levels[52]. The elevated phenylalanine levels may be related to impaired liver or kidney function, as the hydroxylation of phenylalanine to tyrosine occurs in these organs[53]. At the same time, tyrosine levels are reduced in the COVID-19 + AHT cohort, raising further suspicion that this process may be impaired, as it is also the case in patients with severe liver injury[54], and chronic kidney failure[55]. The increased levels of glutamic acid in the COVID-19 patients may result from an increased supply from glutamine, for the synthesis of glutathione as an antioxidant or as an alternative

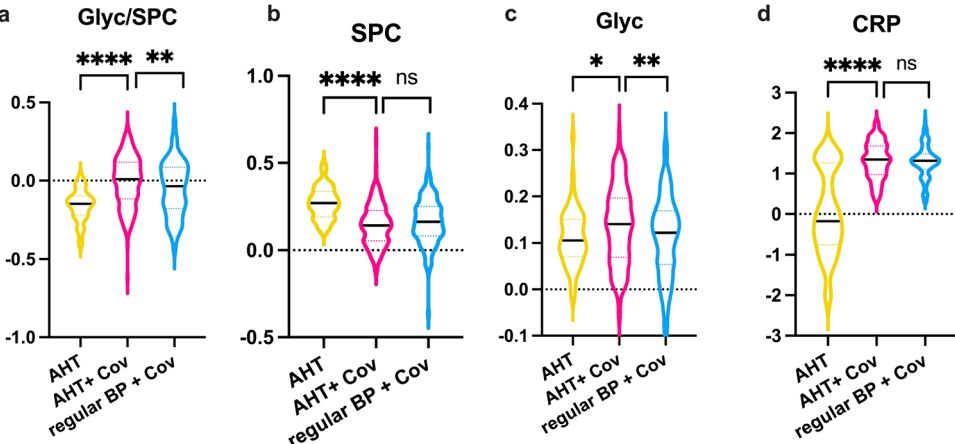

**Fig. 5 COVID-19 patients with AHT show higher inflammatory NMR parameters.** Violin plots with max. to min., whisker, and median, showing the inflammatory NMR parameters Glyc/SPC (**a**), SPC (**b**), Glyc (**c**), and routine labratory CRP (**d**). Unpaired $t$-tests were performed for the comparison of AHT ($n = 58$) versus COVID-19 + AHT ($n = 216$), and COVID-19 with and without AHT ($n = 293$). Significant alterations are shown in form of asterisks: ns (not significant), *$p < 0.05$, **$p < 0.01$, ***$p < 0.001$, ****$p < 0.0001$. The $y$-axis shows the normalized concentration. AHT arterial hypertension, BP blood pressure, Cov COVID-19, CRP C-reactive protein.

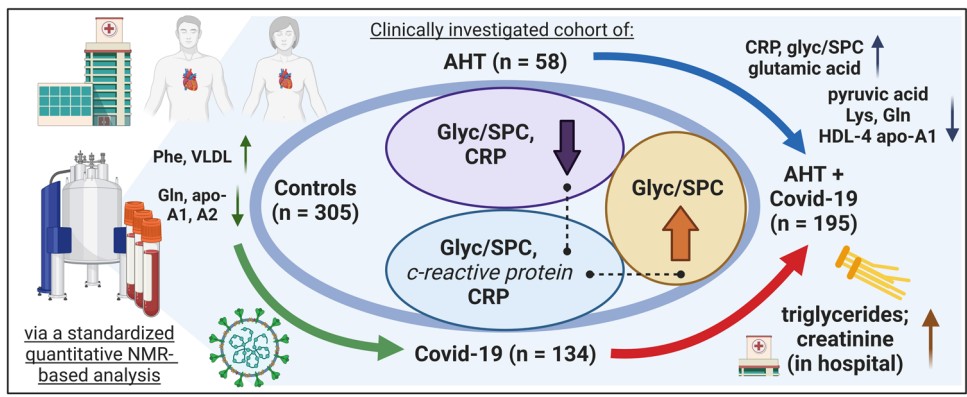

AHT: Arterial HyperTension, CRP: C-Reactive Protein, glyc: glycoprotein, SPC: Supramolecular Phospholipid Composite signal, Lys: Lysine, Gln: Glutamine, HDL: High-Density Lipoprotein, Phe: Phenylalanine, VLDL: Very-Low Density Lipoprotein, apo: apolipoprotein

**Fig. 6 Summary Figure.** This figure is intended to summarize the work process and results of this study. The three cohorts are shown with numbers of participants, as well as the subgroup AHT + COVID-19 with 195 patients. As typical striking metabolic parameters in COVID-19 disease, an increase of CRP, Glyc/SPC and gluatmic acid, as well as a decrease of pyruvic acid, lysine, glutamine, and HDL-4 fractions like HDL-4 apolipoprotein A1 are noted.

energy resource for gluconeogenesis[56]. Furthermore, COVID-19 patients exhibit elevated levels of the Krebs cycle intermediate succinic acid. This is possibly an expression of a hypoxic metabolic state, since succinic acid is able to stabilize hypoxia-inducible factor-1α (HIF-1α)[57] and also to force the interleukin production[58].

Male sex is a known risk factor for severe COVID-19 progression and complications[59,60]. In our sex comparison of COVID-19 individuals, several parameters could be identified as markers for an unfavorable disease course. These included elevated ketone bodies (acetone, acetoacetic acid, 3-hydroxybutyric acid) and marked dyslipidemia[16]. The latter was indicated by increased VLDL particles in males and decreased HDL levels compared to female SARS-CoV-2 infected individuals. From this, one could deduce that the risk factor male sex can also be observed in the metabolome. However, a different picture emerged when the AHT factor was also included in this comparison. Only isolated VLDL particles then showed increased values. Ketone bodies, Glyc/SPC and HDL particles were also no longer found to be significantly altered. The sex differences were mainly related to an increase in various LDL fractions in women, as well as an increase in the BCAAs leucine and isoleucine, citric

acid, and three VLDL particles. The elevated levels of dimethyl-sulfone in our study could be attributed to two patients with very high levels and is possibly the result of a specific diet. However, considering sex and AHT, there were some common features that, although not specific for COVID-19, emerged as typical metabolic profile in with SARS-CoV-2 infected individuals. These included increases in CRP and Glyc/SPC, and decreases in glutamine, isoleucine, leucine, lysine, citric acid, HDL-4 subsets, total cholesterol, and SPC. This is consistent with other metabolomics studies[8,9,13,14] on COVID-19 but gives a finer picture of metabolic characteristics. Some of these metabolic changes in serum have also been found in other viral respiratory infections. One example is the H1N1 influenza virus. Here, decreased levels in the amino acids glutamine and histidine, as well as citric acid were observed[61,62].

Besides male sex, AHT has also previously been cited as a risk factor for severe COVID-19 progression[4] and is also associated with an increased risk of various cardiovascular complications, such as heart failure or acute coronary syndrome[63]. In addition to the already discussed changes in the lipoproteome in cardiac pre-diseased patients, the metabolite trimethylamine-N-oxides (TMAO) should also be mentioned. This metabolite, which is

synthesized by intestinal bacteria, is considered a cardiovascular risk marker due to its proartherosclerotic effects[64].

As expected, also in our cohort with mostly milder courses, hypertensive COVID-19 patients showed an increase in the mentioned cardiovascular risk markers compared with normotensive COVID-19-affected individuals. Interestingly, the ABA1 ratio did not stand out as significantly elevated in hypertensive patients in our cohort. Furthermore, signs of impaired renal function were clearly observed in hypertensive COVID-19 patients. However, these markers are less an expression of COVID-19 but rather due to AHT as such. Thus, when comparing the AHT-cohorts with and without COVID-19, there was no increase in these parameters and thus no further deterioration of renal function during acute infection. However, an indication of renal impairment in the COVID-19 + AHT cohort, is the observation that tyrosine was found to be significantly decreased in hypertensives only in the comparison AHT with and without COVID-19. The triglyceride-rich lipoprotein profile also did not appear to be linked to COVID-19. Succinic acid, which was elevated in hypertensives, could represent a more severe COVID-19 course as an expression of a hypoxic metabolic state[57]. TMAO, which was increased in hypertensive COVID-19 patients, was also associated with a more severe COVID-19 course in patients with pre-existing diabetes mellitus or cardiovascular diseases in one study[65]. Also, COVID-19 patients with pre-existing AHT showed higher inflammatory values in terms of Glyc and Glyc/SPC ratio in our study compared to patients without pre-existing AHT. Comparing hospitalized COVID-19 patients with and without AHT, it was also not surprising that COVID-19 patients with AHT had higher renal values (creatinine, BUN), as well as a lipoprotein profile rich in triglycerides.

With regard to antihypertensive treatment, the use of angiotensin-converting enzyme inhibitors (ACEI) and angiotensin II receptor blockers (ARB) was discussed at the beginning of the pandemic. It was suspected that these classes of drugs, by increasing the expression of ACE-2 receptors, would lead to increased viral uptake into cells and thus to a worse course of COVID-19 disease[66]. However, it has now been shown that treatment with ACEIs and ARBs is not associated with a more severe disease course or increased mortality, but rather the opposite, and blood pressure control is an important factor in COVID-19 treatment[67,68]. Also in our results, COVID-19 + AHT patients, treated with this group of drugs had lower mean levels of renal creatinine and a higher GFR than patients treated with other antihypertensives. However, with the use of RAASI, there was also an increase in some VLDL particles. This could be due to a higher number of multimorbid patients in this group.

In conclusion, we present an NMR-based study examining serum from a large cohort of outpatients and hospitalized COVID-19 patients. Extensive meta-data allowed us to relate to common laboratory parameters, as well as to include in our analyses the important comorbidity AHT, which occurs with high prevalence. This provides a much finer picture of the metabolic fingerprint of COVID-19 disease. Our results might also be expected in other viral respiratory infections because of metabolic similarities in serum. The extensive quantification of metabolites and lipoproteins with an uncomplicated and rapid measurement method such as NMR opens many possibilities in the field of pathophysiology research and risk stratification, which has not yet been conclusively elucidated in COVID-19.

We are aware that the statistical power is diminished by larger differences in cohort sizes and by the fact that no comparisons of laboratory parameters and parameter from the B.I. PACS™ report could be included in the comparison of the COVID-19 patient cohort with the healthy control group from Bruker (HC).

## Data availability

Supplementary Data 2 contains the volcano plot analyses data for the Figs. 2a, b, 3a, b, and 4 from the main manuscript, and for the volcano plots in the Supplementary Figs. 3a, b, 9, and 11. For Fig. 5, and for Supplementary Figs. 1, 2, 4–8, 10, 12, and 13, the NMR Data from Supplementary Data 3 were divided into subgroups for the statistical comparisons using the available metadata for the AHT control cohort (Supplementary Data 4), and the COVID-19 cohort (Supplementary Data 5). Data reports from the applied commercial panels B.I.BioBank-QC™, B.I.Quant-PS™, B.I.LISA™, and B.I. PACS™ are available upon request.

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

## Acknowledgements

We thank the Werner Siemens Imaging Center with the Chair of Department Prof. Dr. Pichler for the opportunity to perform this research. In addition, we greatly thank Karin Tarbet from the Coronataxi-team for her great work and effort and the medical technicians Petra Klöters-Plachky and Alexandra Hof for their biobanking work. Further, we want to thank the members of CRC/TR240 project Z03 (Cornelia Fiessler, Anna Grau, Steffi Jiru-Hillmann, Kirsten Haas) from the institute for clinical epidemiology and biometry, University Hospital Würzburg, Würzburg, Germany and Frederic Emschermann from the Department of Internal Medicine III, Cardiology and Angiology, University Hospital Tübingen, Tübingen, Germany for the good cooperation. We also thank our technical assistants Miriam Owczorz and Daniele Bucci for the support during sample preparation and NMR measurements. This project was supported by the Deutsche Forschungsgemeinschaft (DFG, German Research Foundation)-project number 374031971- TRR 240.

## Author contributions

Conceptualization, C.T. and U.M.; methodology, C.T.; software, C.C. and H.S.; validation, J.K., G.B., and C.T.; formal analysis, J.K., G.B. and C.T.; investigation, J.K., C.T., and U.M.; resources, C.T., T.G., M.G., and U.M.; data curation, J.K. and G.B.; writing—original draft preparation, J.K., C.T. and U.M.; writing—review and editing, J.K., G.B., C.C., H.S., T.G., A.R., M.G., U.M., and C.T.; visualization, J.K. and G.B.; supervision, C.T. and U.M.; project administration, C.T.; funding acquisition, C.T., T.G., M.G., and U.M. All authors have read and agreed to the published version of the manuscript.

## Funding

## Competing interests

C.T. and G.B. report a research grant by Bruker BioSpin GmbH. C.C. and H.S. are employed by Bruker BioSpin GmbH. Their role was in providing healthy control cohort data and were not involved in study design and data analysis. All other authors declare no competing interests.
