## [Peer Review File · Communications Medicine]

Reviewers' comments:

Reviewer #1 (Remarks to the Author):

Dear Editor,

I am writing to express my gratitude for inviting me to review the manuscript "Stratification of hypertensive COVID-19 patients by quantitative NMR spectroscopy of serum metabolites, lipoproteins and inflammation markers – COMMSMED-23-0099." I have carefully evaluated the manuscript and its supplementary data and found it to be innovative, well-written, and effectively addressing the research problem through NMR combined statistical analysis. The authors have done an impressive job of outlining the study's objectives and ensuring that the literature cited is relevant to the study's goal.

However, I have a few recommendations for the authors to consider. Firstly, while the manuscript is well-written, breaking down some of the longer sentences would enhance clarity. Additionally, a thorough check for typos would ensure the manuscript's professional appearance. Secondly, the quality and visibility of "Fig. 2A/B" and "Fig. 4A/B" could be improved. These figures are essential to the manuscript, and enhancing their quality would further improve the manuscript's quality. Lastly, displaying a few NMR spectra in context, if possible, could aid in comprehension for general readers.

I would also like to suggest that the authors correct the superscript "1" in ^1H NMR and other relevant citations in the bibliography where necessary. This would further enhance the manuscript's professional appearance.

Once again, thank you for giving me the opportunity to review this manuscript, and will be happy to review future works for Comm.Med. I appreciate the chance to contribute to the publication process and look forward to reading the final version.

Sincerely,

Reviewer #2 (Remarks to the Author):

The authors present a fundamentally interesting paper with the main question whether the lipoprotein / metabolic signature reported in the blood of COVID-19 patients is actually caused by COVID-19 or by the underlying hypertension.

The same assumption was made by a paper following the preprint of ref 12 which speculates that the signature seen for COVID-19 may be the cause and not a consequence.

Unfortunately it is hard to see the main story line in this manuscript. When looking at the data I would conclude from Fig 2b that only 5 parameters actually COVID-19 related, the rest is AHT. To make things worse, of those by far the most pronounced is the good old CRP. Why the authors then pick pyruvic acid for a univariate representation, I do not understand, as it does not show in the volcano plot above (succinic acid does). The conclusion of this would be that the signature reported in several other papers is not related to COVID-19 but to the underlying AHT.

Some of these changes begin to become very dilute when broken down in female vs male and I

would put those into suppl material.

Also some markers in the more detailed analysis in Fig 5 are questionable - why would dimethylsulfone show changes (this may be a wrong assignment in the software), and univariate plots in 5C show no distinction.

Finally Fig 6 adds little but to show that the hospitalised patients show kidney markers which improve under treatment (Fig 7). This is not worth 2 Figures.

There are too many intermingled stories in this manuscript and it is hence hard to read. If the conclusion is that increased TG subfractions in COVID-19 as reported by several groups just reflect the underlying cause for the disease, then this should be stated clearly, showing all the important uni-variate indicators.

For all of the side-lines the authors should focus on significant differences found which can be underpinned by clear difference in some univariate parameters. The rest should be shown in supplementary material.

In my opinion a condensed manuscript with one or two clear conclusions would be preferable.

Reviewer #3 (Remarks to the Author):

This is a very detailed paper, with a lot of data. I have a few comments

It would be sensible to provide a definition of arterial hypertension given the thrust of the paper is around how those with hypertension and COVID-19 differ from those without hypertension. Was there any data on the severity / duration of hypertension and how it influenced the results?

It should be mentioned that these data are from early in the pandemic and possible changes in findings due to viral evolution (these infections would have been ancestral Wuhan strain or Alpha variant) AND hybrid immunity from prior infection(s) / vaccination, should be acknowledged

The differences in findings by gender are interesting but I note from the supplementary data that both statin and metformin use was much higher in women and may act as a confounder. Is it possible to adjust for medication usage when analysing these data? Given the numbers of statistical tests some correction for multiple comparisons would be sensible.

Reviewer #1 (Remarks to the Author):

I have carefully evaluated the manuscript and its supplementary data and found it to be innovative, well-written, and effectively addressing the research problem through NMR combined statistical analysis. The authors have done an impressive job of outlining the study's objectives and ensuring that the literature cited is relevant to the study's goal. However, I have a few recommendations for the authors to consider.

Response: Thank you very much for these positive comments. This work mainly was possible due to an excellent cohort and corresponding metadata and shall serve for future investigations in hypertensive research beyond COVID-19.

Firstly, while the manuscript is well-written, breaking down some of the longer sentences would enhance clarity. Additionally, a thorough check for typos would ensure the manuscript's professional appearance. Secondly, the quality and visibility of "Fig. 2A/B" and "Fig. 4A/B" could be improved. These figures are essential to the manuscript, and enhancing their quality would further improve the manuscript's quality.

Response: We broke down longer sentences and corrected for typos. We also improved the visibility and resolution of the figures.

Lastly, displaying a few NMR spectra in context, if possible, could aid in comprehension for general readers.

Response: An assigned NMR spectrum was added to the supplement (Supplementary Figure 14).

I would also like to suggest that the authors correct the superscript "1" in ^1H NMR and other relevant citations in the bibliography where necessary. This would further enhance the manuscript's professional appearance.

Response: We corrected the superscript "1" in ^1H NMR throughout the manuscript and formatted the references.

Reviewer #2 (Remarks to the Author):

The authors present a fundamentally interesting paper with the main question whether the lipoprotein / metabolic signature reported in the blood of COVID-19 patients is actually caused by COVID-19 or by the underlying hypertension. The same assumption was made by a paper following the preprint of ref 12 which speculates that the signature seen for COVID-19 may be the cause and not a consequence.

Unfortunately it is hard to see the main story line in this manuscript. When looking at the data I would conclude from Fig 2b that only 5 parameters actually COVID-19 related, the rest is AHT. To make things worse, of those by far the most pronounced is the good old CRP. Why the authors then pick pyruvic acid for a univariate representation, I do not understand, as it does not show in

the volcano plot above (succinic acid does). The conclusion of this would be that the signature reported in several other papers is not related to COVID-19 but to the underlying AHT.

Response: There seems to be ambiguity here on the side of the reviewer regarding the statement of the volcano plots. The misunderstanding has arisen that in Figure 2B the features on the right side, marked with red points are the only 5 COVID-19 related. This is not correct. The aim of this figure is to show how the results of the statistical analyses change when a cohort is subdivided again, in this case based on the pre-disease arterial hypertension (AHT). Thus, Figure 2A shows the comparison of the entire COVID-19 cohort with a healthy cohort, as has often been done in the cited literature. Figure 2B shows only hypertensive COVID-19 patients compared with a hypertensive non-infected comparison group. The factor AHT is thus abbreviated out. It can be seen that more significant metabolic changes are seen in 2A than in Figure 2B, which may well allow the hypothesis that many metabolic changes are actually not caused by COVID-19. Nevertheless, there is overlap. These have been additionally tested for the further metabolic confounder gender using the same procedure (Figure 3) and marked by the red boxes. Accordingly, the COVID-19 typical metabolic changes in our cohort are a reduction in some amino acids (glutamine, lysine, leucine, and isoleucine), citric acid, total cholesterol (TPCH), HDL-4 subgroups (H4A1, H4A2, H4CH, H4PL), and SPC, as well as higher levels of CRP, Glyc/SPC, and glutamic acid. Because laboratory values such as CRP and no Glyc/SPC values from the B.I.PACS report from Bruker were not available for the healthy cohort, a few more features are shown in red boxes in Figure 2B.

In 2C and 2D, as an exemplary representation of individual NMR parameters, the NMR parameters with the highest significant decrease and increase, respectively, were chosen (the y-axis in the volcano plot indicates significance). CRP is not an NMR parameter, but a value from the routine laboratory and was therefore not selected for this representation, but Glyc/SPC. These parameters in 2C and 2D may not show very large differences in the groups, but these differences are still highly significant, as indicated by the asterisks.

Some of these changes begin to become very dilute when broken down in female vs male and I would put those into suppl material.

Response: We have completed the figure descriptions and hope that the statements of the plots are now clearer. We agree that the former Figure 3 does not quite fit into the storyline of the manuscript and have added it as Supplementary Figure 3. We take it as prerequisite for Figure 3 in the main manuscript.

Also some markers in the more detailed analysis in Fig 5 are questionable - why would dimethylsulfone show changes (this may be a wrong assignment in the software), and univariate plots in 5C show no distinction. Finally Fig 6 adds little but to show that the hospitalised patients show kidney markers which improve under treatment (Fig 7). This is not worth 2 Figures.

Response: To formerly Figure 5 (Now Figure 4): The features in 4 c may not show large differences between the cohorts compared. Nevertheless, they show significantly different values in normotensive and hypertensive COVID-19 patients. Further, TMAO and TPTG are considered cardiovascular risk factors, and their importance in this context is addressed in the

Discussion. Therefore, we selected those parameters for better illustration. Towards the assignment of dimethylsulfone: we were using a commercial IVDr based SOP and only used herein identified metabolites (so no proper manual assignment). As we cannot explain the meaning of dimethylsulfone in detail we also didn't discuss the change of this metabolite in detail. When talking about AHT as a risk factor, it makes sense to draw a comparison to normotensive patients. As described in the discussion, this did not show any parameters in the metabolome that were due to COVID-19, but rather commonly known cardiovascular risk factors. The presentation of what could not be proven by NMR in our study is justified in our opinion. We agree that the comparisons between hospitalized normotensive and hypertensive COVID-19 patients is redundant, so this figure (formerly Fig 6) can be found as Supplementary Figure 9. Also in the supplement now is the analysis of antihypertensive treatment groups (Supplementary Figure 11, formerly Fig 7).

There are too many intermingled stories in this manuscript and it is hence hard to read. If the conclusion is that increased TG subfractions in COVID-19 as reported by several groups just reflect the underlying cause for the disease, then this should be stated clearly, showing all the important uni-variate indicators.

Response: Regarding the statement there are “too many intermingled stories” in the manuscript: This manuscript aims to provide a clearer picture of the serum metabolome measured by NMR of COVID-19 patients taking into account a common pre-existing condition such as AHT, as well as gender. Both have been poorly considered in many NMR studies, resulting in certain metabolic alterations being attributed to COVID-19 that were presumably preexisting. These include, for example, the aforementioned elevated TG fractions, which were shown in our study not to be due to COVID-19.

In contrast, our NMR study shows a more accurate metabolic profile with fewer metabolic changes than in the previous and cited literature. It also shows that it is important to pay closer attention to individual characteristics such as previous diseases and gender when examining serum samples by NMR.

For all of the side-lines the authors should focus on significant differences found which can be underpinned by clear difference in some univariate parameters. The rest should be shown in supplementary material.

Response: Regarding the statement “the authors should focus on significant differences”: All parameters shown in Figures 2 to 5 had significant differences for each of the statistical comparisons mentioned. The differences between the compared groups may often not be large. Nevertheless, they are significant differences with a fold change > 1.2 and $p < 0.05$. The same is true for Table 4, which shows all (and only) statistically significantly changed parameters in the respective compared groups.

In my opinion a condensed manuscript with one or two clear conclusions would be preferable.

Response: We agree that a more focused manuscript with less data would be preferable. Therefore, formerly Figure 3, 6, and 7 can be found as Supplementary Figures 3, 9, and 11. We

hope that the manuscript is now more focused and the storyline clear.

Reviewer #3 (Remarks to the Author):

This is a very detailed paper, with a lot of data. I have a few comments

Response: Thank you very much for these positive comments.

It would be sensible to provide a definition of arterial hypertension given the thrust of the paper is around how those with hypertension and COVID-19 differ from those without hypertension. Was there any data on the severity / duration of hypertension and how it influenced the results?

Response: Regarding the definition of AHT: All hypertensive COVID-19 patients were diagnosed with chronic AHT before the COVID-19 pandemic. There was no data of the exact duration and severity of AHT. 120 of the 134 hypertensive COVID-19 patients were under antihypertensive treatment at the time of the study. We are aware that AHT can be classified in three degrees of severity. However, this classification becomes obsolete under treatment. We integrated this statement in the manuscript under the Result section in the beginning under the section "Cohort composition".

It should be mentioned that these data are from early in the pandemic and possible changes in findings due to viral evolution (these infections would have been ancestral Wuhan strain or Alpha variant) AND hybrid immunity from prior infection(s) / vaccination, should be acknowledged

Response: The data were from the early phase of the pandemic, when the alpha virus or ancestral Wuhan strain was the predominant viral variant. Therefore, changes in findings due to viral evolution are possible. All patients were primarily affected. During the period between January and May 2021, six of the 329 COVID-19 patients included in this study (after quality control) were vaccinated with the vaccines from BionTech, AstraZeneca and Moderna, respectively. We integrated this statement in the manuscript in the Material and Methods section under the section "patient cohort and sample collection".

The differences in findings by gender are interesting but I note from the supplementary data that both statin and metformin use was much higher in women and may act as a confounder. Is it possible to adjust for medication usage when analysing these data? Given the numbers of statistical tests some correction for multiple comparisons would be sensible.

Response: Concerning the medication (statin and metformin) and gender distribution, we would like to note that of the 46 study participants from the COVID-19 cohort taking statins, 13 were female and 33 were male. Of the 20 study participants taking metformin, 7 were female and 13 were male. The sex ratio for metformin was thus almost balanced, while that for statins was more on the side of males. For multiple comparisons we corrected with false discovery rate (FDR < 0.01). This applies to all statistical tests and therefore to Table 4, as well as to Figures 2 to 5. We have highlighted this again in the Methods section under the Statistical section.

REVIEWERS' COMMENTS:

Reviewer #1 (Remarks to the Author):

None

Reviewer #2 (Remarks to the Author):

This manuscript can now be accepted.

Reviewer #3 (Remarks to the Author):

Thank you to the authors for addressing my comments. It is a shame no details on the duration / extent of hypertension were available but I understand why this is and have no further comments